# A Context Semantic Auxiliary Network for Image Captioning

Jianying Li [1,2] and Xiangjun Shao [1,3,*]

1. School of Computer and Electrical Engineering, Hunan University of Arts and Science, Changde 415000, China; ljymnn@huas.edu.cn
2. Key Laboratory of Hunan Province for Control Technology of Distributed Electric Propulsion Air Vehicle, Changde 415000, China
3. School of Computer Science, Wuhan University, Wuhan 430072, China
* Correspondence: shaoxiangjun@whu.edu.cn

**Abstract:** Image captioning is a challenging task, which generates a sentence for a given image. The earlier captioning methods mainly decode the visual features to generate caption sentences for the image. However, the visual features lack the context semantic information which is vital for generating an accurate caption sentence. To address this problem, this paper first proposes the Attention-Aware (AA) mechanism which can filter out erroneous or irrelevant context semantic information. And then, AA is utilized to constitute a Context Semantic Auxiliary Network (CSAN), which can capture the effective context semantic information to regenerate or polish the image caption. Moreover, AA can capture the visual feature information needed to generate a caption. Experimental results show that our proposed CSAN outperforms the compared image captioning methods on MS COCO "Karpathy" offline test split and the official online testing server.

**Keywords:** deep learning; attention mechanism; image captioning

## 1. Introduction

Image captioning is a challenging task in the field of artificial intelligence, as it involves generating coherent and natural language sentences that describe an input image. This task serves as a bridge between computer vision and natural language processing, and has gained considerable attention in recent years.

The majority of existing image captioning methods follow the encoder–decoder framework [1–14], where the encoder first employs a convolutional neural network (CNN) to extract visual features of an input image and the decoder mainly utilizes a recurrent neural network (RNN) to generate a descriptive sentence for the given image. Later, attention mechanisms [15] were introduced in the encoder and decoder, which were firstly applied in machine translation, and achieved great improvements in image captioning. With the advance of Transformer, many transformer-based architecture models [16–18] were proposed and great improvements made in image captioning. For example, DLCT [16] explores the intrinsic properties of region and grid features for image captioning. However, these captioning methods [19–22] predict the current word depending on the previously generated words, which results in a lack of context semantic information in the model, as illustrated in Figure 1.

In recent years, several works have used contextual semantic information to generate image captions. RD (Rumination decoding) [23] draws on the practice of people modifying and polishing articles after writing, and proposes a second polishing of image description sentences. It first uses the basic decoder to generate a rough descriptive sentence, and then corrects and polishes the generated description sentence. CAAG (Context-Aware Auxiliary Guidance) [24] divides the decoding process into two stages. In the first stage, a module, called the basic decoder, is used to generate a rough description sentence. Conditioned on the sentences generated in the previous stage, the second stage uses a decoder to regenerate

more accurate sentences. Experimental results of CAAG show that it significantly improves the performance of image description models.

Although RD and CAAG effectively leverage contextual semantic information, which proves useful for certain tasks such as visual segmentation [25], there are still limitations associated with them. Specifically, RD and CAAG initially generate complete sentences that may contain unavoidable errors or irrelevant information. In subsequent image captioning processes, the language model with attention mechanism regenerates the caption sentences. During this regeneration, the attention mechanism assigns weights to both the erroneous and irrelevant information, thereby constraining the quality of the secondary generated descriptive sentences.

To address this issue mentioned above, we propose Attention-Aware (AA), which extends the conventional attention mechanism [15] by adding a gated nonlinear unit. Specifically, AA can generate an "information vector" with a linear transformation and a nonlinear activation, which is similar to GTU (Gated Tanh Unit) [26], and also generate a gate vector with a linear transformation and another nonlinear activation. Based on AA, we propose a Context Semantic Auxiliary Network (CSAN), which mainly combines the proposed AA mechanism with LSTMs to generate the description sentence of the input image. Specifically, like CAAG, we first use the captioning model (called the base network) to generate a complete descriptive sentence serving as context semantic information. And then, CSAN uses the AA mechanism to filter out erroneous or irrelevant context information. Moreover, CSAN also selectively and effectively decodes visual features for regenerating the image caption. Figure 1 shows the comparison between the traditional method and our proposed method.

To evaluate our proposed method, we performed comprehensive experiments using the publicly available image caption dataset MS COCO. Furthermore, to assess the adaptability of our approach, we applied it to extend three well-known image captioning methods, namely BUTD [27], RDN [28], and AoA-Net [29]. The experimental results demonstrate significant improvements achieved by our proposed method across various metrics.

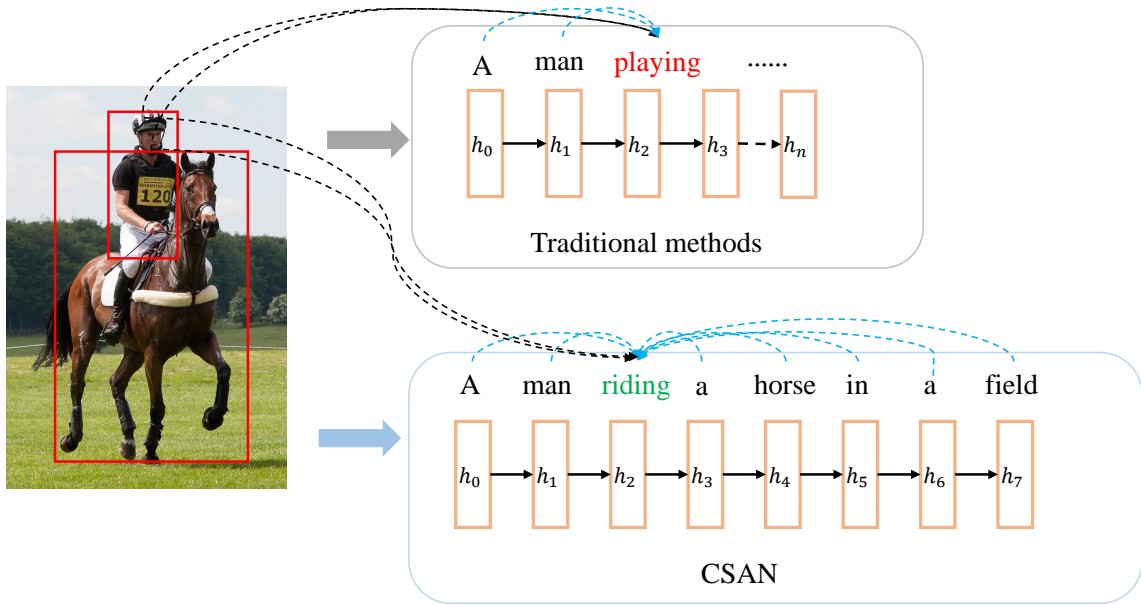

**Figure 1.** Traditional method and our proposed method. The red font refers to the prediction from the previous predicted words and the green font refers to the prediction contextual information.

To sum up, the main contributions of our proposed method are listed as follows:

- We propose an Attention-Aware (AA) mechanism, which can effectively filter out erroneous or irrelevant information.
- A Context Semantic Auxiliary Network (CSAN) is constituted via AA, which can effectively capture the context semantic information from the complete descriptive sentence, and results in great improvement.

## 2. Related Works

### 2.1. Image Captioning

Earlier image captioning methods are mainly template-based [30–32] with fixed templates with slots, which use the detected objects, predicted attributes prediction, and recognized scene to fill in the slots. Inspired by neural machine translation [33], recent image captioning methods mostly follow the encoder–decoder framework [9,27–29]. Specifically, the encoder utilizes CNN to extract visual features of an input image, and the decoder uses a language generation model to generate a complete description sentence of the image. For instance, ref. [1] first proposed the encoder–decoder framework in image captioning, using a CNN to extract image features and an RNN to generate caption sentences by decoding the extracted image features. With the significant development of attention mechanism, ref. [14] used an attention mechanism to attend the spatial area for captioning and made a significant improvement in image captioning. VAR (Visual Abductive Reasoning) [34] explored abductive reasoning to capture the context from visual premise information for generating an image description. Several works have mined the context semantic information to generate the caption sentence. RD [23] first explored the contextual information to generate a descriptive sentence for a given image. CAAG [24] learned the global information to guide generation of the image caption and greatly enhanced the performance of the caption model.

### 2.2. Attention Mechanism

The attention mechanism [15] was initially applied to machine translation and achieved great improvements. It first calculates a dependency value for each candidate feature vector and then uses the softmax function to normalize the calculated dependency values to weights, finally applying these weights to the candidates to generate a weighted average vector. Based on this, other attention mechanisms have been proposed, for instance, adaptive attention [35], multi-level attention [36], self-attention and multi-head attention [37]. Adaptive attention with a sentinel learns whether to capture the image information or the sentinel for word prediction. Self-attention captures the relationship among the region features of a given image, which greatly improves the quality of the image caption. Multi-head attention extends self-attention via multi-head, which remarkably enhances the performance of the machine translation. Unlike these, our proposed AA mechanism extends the traditional attention via a gate mechanism for filtering out undesired information for image captioning.

## 3. Methods

This section provides a detailed introduction to our proposed image captioning method called CSAN, depicted in Figure 2. The method consists of three key steps. Firstly, it utilizes a convolutional neural network (CNN) to extract visual features from the input image. These features capture important visual information. Secondly, a base network, referred to as the image captioning model, is employed to generate a complete caption sentence for the input image. The base network leverages the visual features obtained from the CNN to generate descriptive sentences. Lastly, CSAN generates a new caption sentence for the same input image, conditioned on the previously generated descriptive sentence using the base network. This conditional generation process enables CSAN to produce refined and improved caption sentences.

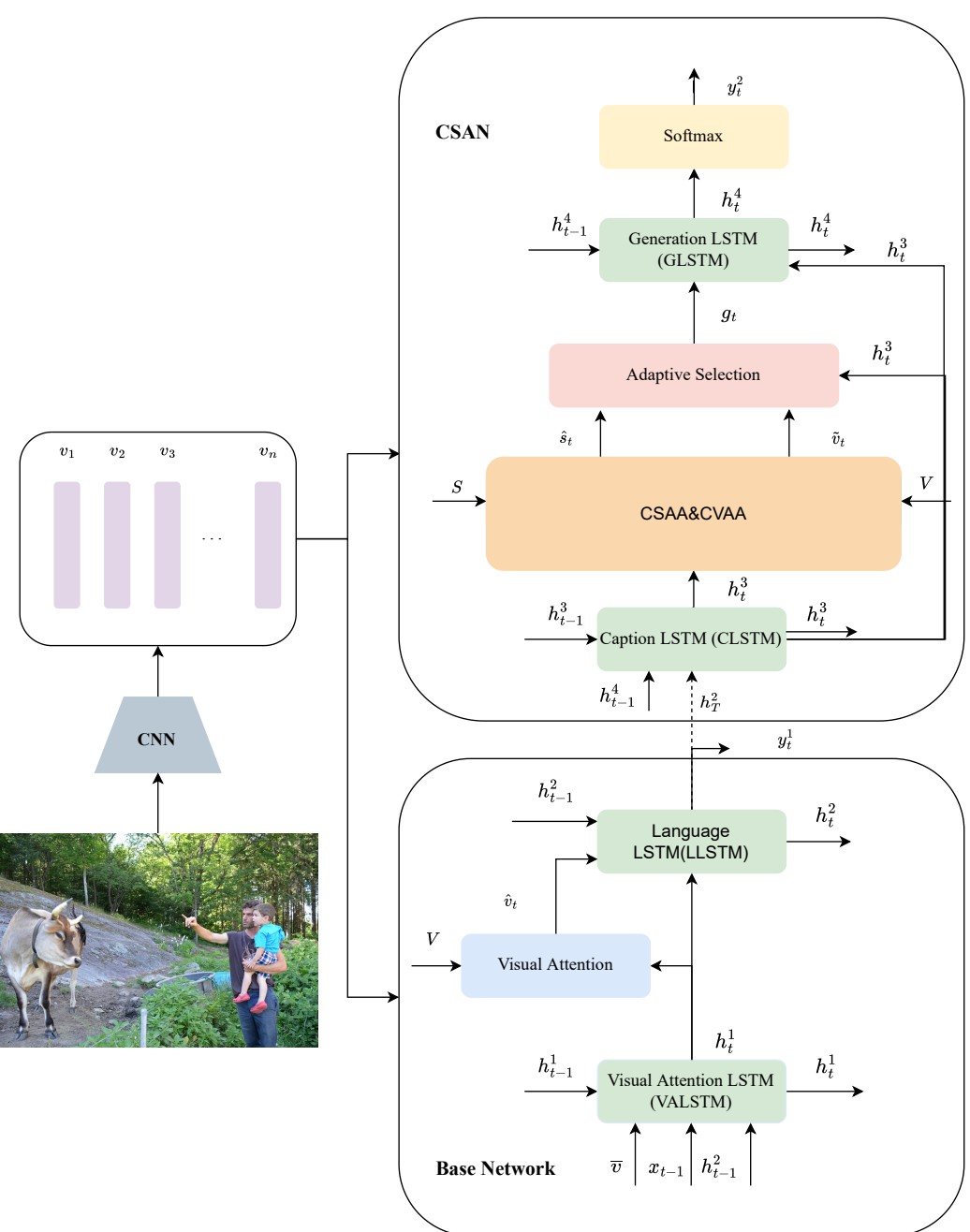

**Figure 2.** The overall workflow of our proposed method. Firstly, CNN is employed to extract the visual features of the input image. Secondly, the base network generates a caption sentence based on the visual features. Finally, CSAN regenerates the sentence with context from the base network and visual features, where CVAA and CSAA use the AA process for visual and context semantic information, respectively.

### 3.1. Visual Feature Extraction

Similar to many captioning methods, CSAN employs the pre-trained Faster R-CNN [38] model to detect objects within the image. Subsequently, it utilizes a pre-trained convolutional neural network model, specifically ResNet101 [39], to extract the visual features from the image. The visual feature extraction process is illustrated below:

$$V = CNN(I), \tag{1}$$

where $V = \{v_1, v_2, v_3, \cdots, v_N\}$ is the visual feature sequence and $N$ is the number of the detected objects in the input image.

### 3.2. Base Network

To provide contextual semantic information for CSAN to regenerate the caption sentence, this section uses the classic image captioning model BUTD [27] as the base network. It is mainly composed of Visual Attention LSTM (VALSTM), language LSTM (LLSTM), and a top-to-bottom visual attention module (Visual Attention) shown in Figure 2.

After extracting the visual features $V = \{v_1, v_2, v_3, \cdots, v_N\}$, the base network first uses VALSTM to get its current state vector serving as a query vector for top-down attention. Then, under the action of top-down attention and query vector, LLSTM with a softmax layer generates image caption sentence to provide contextual semantic information for the subsequent regeneration. The specific process description is shown as follow:

$$h_t^1 = VALSTM(\bar{v}, h_{t-1}^2, x_{t-1}, h_{t-1}^1), \tag{2}$$

where $h_t^1$ is the output of the first layer LSTM (VALSTM) at $t$ time step, $\bar{v}$ is calculated via mean pooling $\bar{v} = \frac{1}{N} \sum_{i=1}^{N} v_i$, and $x_{t-1}$ is the output of the previous step.

$$h_t^2 = LLSTM(h_t^1, \hat{v}_t, h_{t-1}^2), \tag{3}$$

$$p(y_t | y_{<t-1}) = softmax(W_{bp} h_t^2 + b_{bp}), \tag{4}$$

where $h_t^2$ is the output of the second layer LSTM (VALSTM) at $t$ time step, $\hat{v}$ is the weighted vector, calculated as follows:

$$\hat{v}_t = \sum_{i=1}^{N} \alpha_{it} v_i, \tag{5}$$

$$m_{it} = W_{mv} tanh(W_{mh} h_t^1 + W_{mv} v_i + b_m), \tag{6}$$

$$\alpha_t = softmax(m_t), \tag{7}$$

where $W_{mv}$ and $W_{mh}$ are learnable parameters, and $b_m$ is a learnable bias parameter.

### 3.3. Context Semantic Aware Network

The generated sentence from the base network, while providing contextual semantic information for caption regeneration, may contain unavoidable errors or irrelevant information. These shortcomings directly impact the quality of subsequent caption regeneration. To address this challenge, we propose the Context Semantic Auxiliary Network (CSAN) as a solution. In contrast to CAAG, CSAN effectively filters out errors or irrelevant information from the contextual semantic data, enabling the generation of improved image captions. Additionally, CSAN adaptively utilizes visual features to further enhance the quality of the generated captions. By incorporating these strategies, CSAN aims to generate high-quality captions that are more accurate and contextually relevant.

#### 3.3.1. Attention-Aware Module

AA module employs a GTU to extend the traditional attention mechanism, which can filter out the erroneous or irrelevant information. Figure 3 illustrate the traditional attention mechanism shown in Figure 3a and the proposed AA module shown in Figure 3b.

Given a hidden state vector $h_t$ and a vector sequence $V$, AA first computes a weighted average vector, which is calculated as follows:

$$\alpha_{ti} = align(h_t, v_i), \tag{8}$$

$$align(h_t, v_i) = \frac{exp(s(h_t, v_i))}{\sum\limits_{i=1}^{N}(s(h_t, v_i))}, \tag{9}$$

$$s(h_t, v_i) = V_s tanh(W_s[h_t; v_i] + b_s), \tag{10}$$

$$c_t = \sum_{i=1}^{N} \alpha_{ti} v_i, \tag{11}$$

where $align(\cdot)$ is a correlation score function, $V_s$, $W_s$ is the learnable weight parameter, $b_s$ is the learnable bias parameter, and $c_t$ is the global vector derived from vector sequence $V$ and $h_t$.

Subsequently, the vector combined with $h_t$ computes a gate vector $c_g$ via a linear transformation and gate mechanism. Meanwhile, this vector is transformed as another vector $c_v$ via another linear and nonlinear transformation $tanh(\cdot)$. Finally, AA operates on $c_g$ and $c_v$ to get $\hat{c}_t$. These operations are formulated as follows:

$$c_g = Gate(W_g[h_t; c_t] + b_g), \tag{12}$$

$$c_v = V_v tanh(W_v[h_t; c_t] + b_v), \tag{13}$$

$$\hat{c}_t = c_g \odot c_v, \tag{14}$$

where $Gate(\cdot)$ is the *Sigmoid* function, $W_g$ is the learnable weight parameter, $b_g$ is the learnable bias parameter, $V_v$, $W_v$ are the learnable weight parameters, $b_v$ is the learnable bias parameter, and $\odot$ refers to element-wise multiplication.

3.3.2. Context Semantic Aware Network

To leverage contextual semantic information from the base network effectively and mitigate the issue of deviation in image captioning resulting from relying solely on semantic information, we introduce the Context Semantic Auxiliary Network (CSAN) based on the Attention-Aware mechanism described earlier. This network is designed to effectively utilize both contextual semantic information and visual features in generating accurate image descriptions. By incorporating both sources of information, CSAN aims to strike a balance and ensure that the generated captions are not solely reliant on semantic information, thus improving the overall quality and relevance of the image descriptions.

The proposed CSAN consists of two attention-aware modules and two layers of LSTM, as shown in Figure 2. Specifically, the network embeds two attention-aware modules, the Contextual Semantic Attention-Aware module (CSAA) and the Visual Attention-Aware module (CVAA), which are AA module applied in context semantic information and visual features, in parallel between the Context LSTM (CLSTM) and the Generation LSTM (GLSTM). First, the CLSTM module in the network provides query vectors for the two global attention modules. Then, this module selects effective contextual semantic information and visual information based on the query vectors. Finally, the GLSTM generates image descriptions based on the adaptively selected contextual semantic information and visual information.

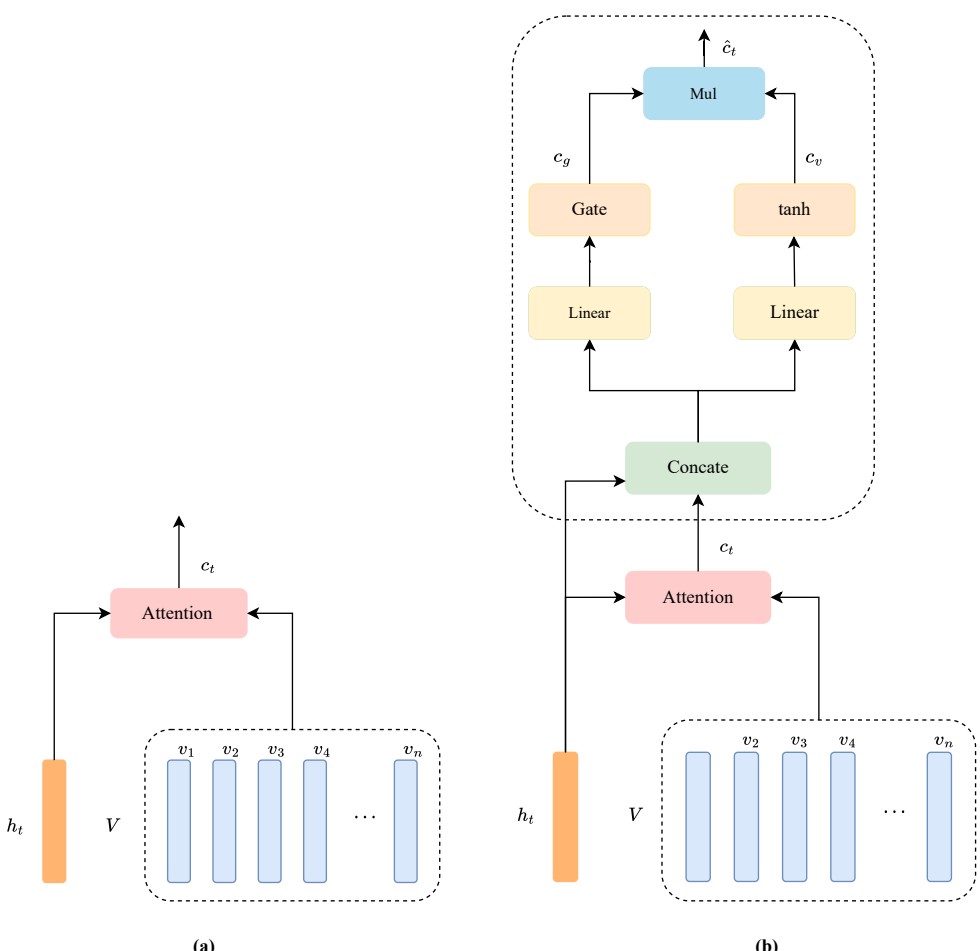

**Figure 3.** Attention and Attention-Aware (AA); (**a**) is the conventional attention mechanism; (**b**) is the AA module which can filter out irrelevant or incorrect information.

First, under the condition of $h_T^2$, the output of $CLSTM$ is $h_t^3$, which is calculated as follows:

$$h_t^3 = CLSTM(h_T^2, h_{t-1}^4, h_{t-1}^3). \tag{15}$$

Then, the output $h_t^3$ of $CLSTM$ is used as the query vector in $CSAA$ and $CVAA$. $CSAA$ is used to filter out erroneous or irrelevant context semantic information, while $CVAA$ is used to select the input visual features. $h_T^2$ is the last state vector of $CLSTM$ generated during the image captioning throughout the base network.

Next, CSAA and CVAA learn the semantic information $\hat{s}t$ and visual information $\widetilde{v}t$ based on the state $h_t^3$ at time $t$, respectively, which is calculated as follows:

$$\hat{s}_t = CSAA(S, h_t^3), \tag{16}$$

$$\widetilde{v}_t = CVAA(V, h_t^3), \tag{17}$$

where $S$ is the word vector sequence generated by the base network, $V$ is the visual feature sequence, and $h_t^3$ is the state vector of $GLSTM$ at the current time $t$.

Subsequently, the context semantic vector $\hat{s}_t$ and visual information $\widetilde{v}_t$ are adaptively selected based on the current state, as shown in Figure 4, which is calculated as follows:

$$g_t = z_t \odot \hat{s}_t + (1 - z_t) \odot \widetilde{v}_t, \tag{18}$$

where $z_t$ is formulated as follows:

$$z_t = sigmoid(Linear(\hat{s}_t, h_t^3)), \tag{19}$$

where $sigmoid(\cdot)$ is an activation function.

Finally, the prediction is generated by GLSTM with softmax, which is formulated as follows:

$$h_t^4 = CLSTM(g_t, h_{t-1}^4, h_t^3), \tag{20}$$

$$y_t^2 = softmax(h_t^4), \tag{21}$$

where $y_t^2$ is the last prediction of CSAN at time $t$.

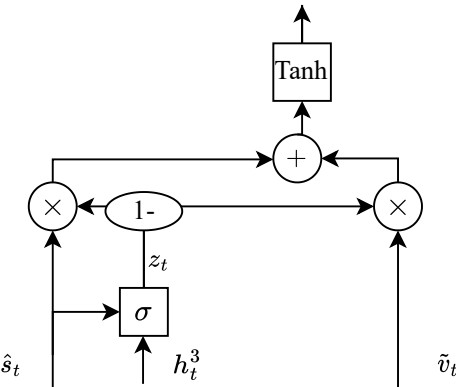

**Figure 4.** Adaptive selection of context semantic information and visual information.

*3.4. Training*

Both the base network and CSAN are trained using minimum cross-entropy loss, described as follows:

$$L_B(\theta_B) = \sum_{t=0}^{T-1} log(p_b(y_t|y_{<t-1}), \tag{22}$$

$$L_G(\theta_G) = \sum_{l=0}^{L-1} log(p_b(y_l|y_{<l-1}). \tag{23}$$

During the training phase, the parameters of the base network and CSAN are updated simultaneously to maximize the objective function as follows:

$$L_{(\theta_B, \theta_G)} = L_B(\theta_B) + L_G(\theta_G). \tag{24}$$

**4. Experiments**

*4.1. Dataset*

To evaluate the effectiveness of our proposed model, extensive experiments were conducted on the dataset MS COCO [40]. This popular dataset contains 123,287 images annotated with five sentences for each, including 82,783 images for training and 40,504 images for validating. The offline "Karpathy" split was used for comparisons, including 5000 images for training, 5000 for validating, and the rest for training. All sentences were converted to lower case and the words that occurred less than five times were dropped. Five different metrics were used for the evaluation of our model, including BLEU-N [41], METEOR [42], ROUGE-L [43], CIDEr [44], and SPICE [45].

### 4.2. Experiment Settings

In this paper, the dropout rate was set to 0.5, the batch size for training was 16, and the number of iterations for training was 60. During training, the model used the Adam optimization algorithm to update the network parameters, with an initial learning rate of $10^{-5}$; the momentum and decay were set to 0.8 and 0.999, respectively. During validation and testing, the beam search algorithm was used to improve the effectiveness of generating the image caption. In this paper, the beam size was set to 3.

### 4.3. Quantitative Analysis

Table 1 provides the evaluation results of CSAN when extended to BUTD, RDN, and AoA-Net, along with comparisons to other benchmark methods. In line with CAAG [24], BUTD was selected as the baseline model, and both models were trained using cross-entropy loss.

From Table 1, it can be observed that BUTD directly utilized context semantic information through the CCAG extension. The performance of BUTD was significantly improved, confirming the importance of context semantic information in image captioning. Similarly, the CSAN extension of BUTD utilized the proposed Attention-Aware mechanism to employ context semantic information and achieved a large performance boost. In addition, this section also extends CSAN to RDN and AoA-Net. The performance of the extensions was also greatly improved. For example, in BLEU-4, the three extended models improved by 2.4%, 2.0%, and 0.5%, respectively, while, in CIDEr, they improved by 16.5%, 14.2%, and 3.7%. Compared with similar extension models such as CAAG, CSAN achieved better results by effectively utilizing context semantic information through the Attention-Aware mechanism and adapting to visual features to generate an image caption. By utilizing context semantic information, the proposed method optimized and polished the first generated image caption, significantly improving the model's performance. Therefore, it can be concluded that the proposed method utilizing the Attention-Aware mechanism can effectively learn context semantic information, which provides essential semantic information for generating an accurate image caption. Furthermore, by adaptively utilizing visual features, the proposed method can effectively enhance the performance of the image description model, generating high-quality sentences. From Table 1, the performance of BUTD, RDN, and AoA-Net increased sequentially, and their performance after CSAN expansion was also sequentially improved. Therefore, it can be seen that higher-quality description sentences can provide more effective semantic information.

To further evaluate the performance of CSAN, we conducted experiments on the Flickr30k dataset. The experimental results of CSAN on this dataset are presented in Table 2. In this section, CSAN is extended to two baseline models: Adaptive and BUTD. The table showcases the test results of CSAN on the Flickr30k dataset, comparing them with other reported image captioning methods on the same dataset.

The results demonstrate that extending CSAN significantly improves the performance of the baseline models. Specifically, the METEOR score improves by 1.9% and 1.3% for the Adaptive and BUTD models, respectively. Similarly, the B-4 evaluation metric improves by 1.2% and 2.3% for the two models, respectively. When CSAN is applied to the BUTD model, it effectively leverages the contextual semantic information provided by BUTD, resulting in superior performance across all evaluation metrics compared to the comparison methods.

**Table 1.** A performance comparison between CSAN and the comparison methods on the dataset MS-COCO "Karpathy" test split.

| Metric | B-1 | B-2 | B-3 | B-4 | METEOR | ROUGE-L | CIDEr | SPICE |
|---|---|---|---|---|---|---|---|---|
| BUTD [27] | 79.0 | 60.4 | 47.8 | 36.3 | 27.0 | 56.4 | 113.5 | 20.3 |
| AoA-Net [29] | 80.2 | - | - | 39.1 | 29.2 | 58.8 | 129.8 | 22.3 |
| GCN-LSTM [12] | 77.4 | - | - | 37.1 | 28.1 | 57.2 | 117.1 | 21.4 |
| RDN [28] | 77.5 | 61.8 | 47.9 | 36.8 | 27.2 | 56.8 | 115.3 | 20.5 |
| SAC [46] | 77.2 | - | - | 36.8 | 28.0 | 57.1 | 116.3 | 21.2 |
| Attin + RD [23] | - | - | - | 36.8 | 28.1 | 57.5 | 116.5 | 21.2 |
| BUTD + CAAG [24] | - | - | - | 38.4 | 28.6 | 58.6 | 128.8 | 22.1 |
| Multi-gate [17] | 78.4 | 62.8 | 48.9 | 37.5 | 28.2 | 57.8 | 117.5 | 21.6 |
| ASIA [47] | 78.5 | 62.2 | 48.5 | 37.8 | 27.7 | - | 116.7 | - |
| CA-VNP [48] | - | - | - | 38.6 | 28.3 | 58.5 | 125.0 | 22.1 |
| AAT [49] | 78.6 | - | - | 38.2 | 29.2 | 58.3 | 126.3 | 21.6 |
| BUTD + CSAN | 79.1 | 63.0 | 49.5 | 38.7 | 28.8 | 59.1 | 130.0 | 22.2 |
| RDN + CSAN | 79.5 | 63.2 | 49.8 | 38.8 | 29.0 | 59.7 | 129.5 | 22.3 |
| **AoA-Net + CSAN** | **80.3** | **64.5** | **52.5** | **39.6** | **29.4** | **60.0** | **133.5** | **22.6** |

**Table 2.** A performance comparison between CSAN and the comparison methods on the dataset Flickr30k.

| Metric | B-1 | B-2 | B-3 | B-4 | METEOR | ROUGE-L | CIDEr | SPICE |
|---|---|---|---|---|---|---|---|---|
| Soft-Attn [14] | 66.7 | 43.4 | 28.8 | 19.1 | 18.5 | - | - | - |
| Adaptive [35] | 67.7 | 49.4 | 35.4 | 25.1 | 20.4 | - | 53.1 | - |
| BUTD [27] | 76.4 | - | - | 27.3 | 21.7 | 56.6 | - | - |
| DAIC [50] | 64.5 | 46.4 | 33.5 | 24.3 | 20.4 | 46.7 | 61.6 | - |
| ARL [51] | 69.8 | 51.7 | 37.8 | 27.7 | 21.5 | 48.5 | 57.4 | - |
| cLSTM-RA [52] | 70.5 | 52.5 | 37.6 | 27.1 | 21.9 | 49.4 | 57.7 | - |
| Trans-KG [53] | 78.4 | - | - | 26.8 | 21.7 | - | 56.6 | - |
| LGVIA [54] | 75.4 | 57.6 | 39.0 | 28.2 | 25.4 | 53.7 | 58.0 | - |
| Adaptive + CSAN | 71.6 | 54.3 | 39.1 | 26.3 | 22.3 | 51.5 | 60.2 | 17.6 |
| **BUTD + CSAN** | **77.3** | **59.2** | **44.3** | **29.6** | **23.0** | **59.8** | **68.7** | **20.7** |

Furthermore, Table 2 highlights that utilizing contextual semantic information in image captioning confers a significant advantage over generating captions solely based on visual features. This indicates that the context semantic information learned through the Attention-Aware mechanism can effectively enhance the quality of the generated sentences.

In addition, we also conducted a comparison of CSAN with the other benchmark methods on the official online test set of MS COCO. The results of this comparison are presented in Table 3. The comparison clearly demonstrates that our proposed captioning method, which incorporates context semantic information, outperforms the other image captioning methods. This further validates the effectiveness of the Attention-Aware mechanism in improving the performance of the image captioning model by effectively mining and utilizing contextual semantic information.

*4.4. Ablation Analysis*

We extensively explored different structures and settings of our method to gain insights into how and why it works. To this end, this section set up the CSAN as follows:

- CSAA and CVAA were removed from CSAN, leaving only two layers of LSTM, which was referred to as the base network.
- CSAA was added to the base network (base + CSAA) to verify the necessity of learning contextual semantic information by CSAA in improving the image captioning model.
- Based on the second step, CVAA was added to the base network (base + CSAA + CVAA) to verify the impact of visual feature information under the condition of context semantic information.

**Table 3.** Comparison of the results of CSAN and the comparative method on the MS-COCO official server.

| c5 | | | | | | | | |
|---|---|---|---|---|---|---|---|---|
| Model | B-1 | B-2 | B-3 | B-4 | METEOR | ROUGE-L | CIDEr | SPICE |
| Adaptive [35] | 74.8 | 58.4 | 44.4 | 33.6 | 26.4 | 55.0 | 104.2 | 19.7 |
| ReviewNet [55] | 72.0 | 55.0 | 41.4 | 31.3 | 25.6 | 53.3 | 96.5 | 18.5 |
| BUTD [27] | 80.2 | 64.1 | 49.1 | 36.9 | 27.6 | 57.1 | 117.9 | 21.5 |
| RF-Net [56] | 80.4 | 64.9 | 50.1 | 38.0 | 28.2 | 58.2 | 122.9 | - |
| RDN [28] | 80.2 | - | - | 37.3 | 28.1 | 57.4 | 121.2 | - |
| AoA-Net [29] | 81.0 | 65.8 | 51.5 | 39.6 | 29.3 | 58.9 | 126.9 | 21.7 |
| HTC [57] | 80.2 | 64.8 | 51.0 | 38.5 | 28.6 | 58.4 | 124.2 | - |
| GAT [58] | 81.1 | 66.1 | 51.8 | 39.8 | 29.1 | 59.1 | 127.8 | - |
| CA-VPN [48] | 81.6 | 64.3 | 50.8 | 37.9 | 27.4 | 57.6 | 120.9 | - |
| **Ours** | 80.7 | **66.0** | **52.6** | **41.0** | **30.9** | **61.2** | **130.4** | **22.7** |
| c40 | | | | | | | | |
| Model | B-1 | B-2 | B-3 | B-4 | METEOR | ROUGE-L | CIDEr | SPICE |
| *Adaptive* [35] | 92.0 | 84.5 | 74.4 | 63.7 | 35.9 | 70.5 | 105.9 | 67.3 |
| ReviewNet [55] | 90.0 | 81.2 | 70.5 | 59.7 | 34.7 | 68.6 | 96.9 | 64.9 |
| BUTD [27] | 95.2 | 88.8 | 79.4 | 68.5 | 36.7 | 72.4 | 120.5 | 71.5 |
| RF-Net [56] | 95.0 | 89.3 | 80.1 | 69.2 | 37.2 | 73.1 | 125.1 | - |
| RDN [28] | 95.3 | - | - | 69.5 | 37.8 | 73.3 | 125.2 | - |
| AoA-Net [29] | 95.2 | 89.6 | 81.3 | 70.9 | 38.6 | 74.9 | 129.6 | 72.6 |
| HTC [57] | 95.1 | 89.0 | 81.2 | 70.4 | 38.4 | 73.6 | 128.9 | - |
| GAT [58] | 95.1 | 89.7 | 81.5 | 71.4 | 38.4 | 74.7 | 130.8 | - |
| CA-VPN [48] | 95.6 | 87.8 | 80.3 | 69.5 | 37.3 | 73.7 | 124.5 | - |
| **Ours** | 95.1 | 89.3 | **82.7** | **73.0** | **40.7** | **76.5** | **134.5** | **73.4** |

According to the data analysis of the three different settings on various evaluation metrics in Table 4, we can seen that, after CSAA was added to the base network, performance of the model significantly improved on all metrics in Table 4. Specifically, context semantic information that CSAN learned led to improvements by 5.2%, 2.8%, 2.4%, 5.9%, 9.6%, and 1.3% in B-1, B-4, METEOR, ROUGE-L, CIDEr, and SPICE, respectively. These results indicate that CSAA can learn valuable context semantic information from image descriptions and significantly improve the performance of CSAN. In addition, unlike CCAG [24], CSAN also adaptively selects and applies visual features based on context semantic information and the current state, which alleviate the bad influence due to the absence of visual features. Table 4 shows that, conditioned on context semantic information, the effective selection and utilization of visual features through CVAA can also further improve the performance of CSAN.

**Table 4.** Comparison of performance under different settings in CSAN.

| Settings/Metrics | B-1 | B-4 | METEOR | ROUGE-L | CIDEr | SPICE |
|---|---|---|---|---|---|---|
| base | 73.2 | 35.6 | 26.0 | 52.8 | 118.8 | 20.8 |
| base + CSAA | 78.4 | 38.4 | 28.4 | 58.7 | 128.4 | 22.1 |
| **base + CSAA + CVAA** | **79.1** | **38.7** | **28.8** | **59.1** | **130.0** | **22.2** |

*4.5. Qualitative Analysis*

Figure 5 shows some examples with images and captions generated by BUTD, CAAG, and CSAN, and conducts in-depth qualitative analysis.

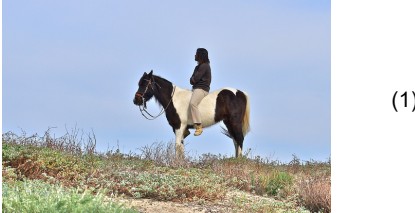

(1)

**Baseline:** one person and a horse are on the ground.

**CAAG:** a person is riding a horse on the ground.

**Ours:** one person is riding a horse on the grass land.

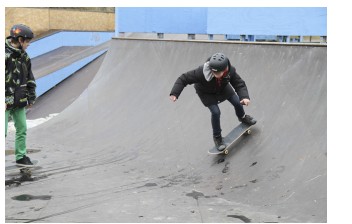

(2)

**Baseline:** a boy is standing on the skateboard.

**CAAG:** a boy is standing on the skateboard.

**Ours:** A boy is playing skateboard sliding down the slope.

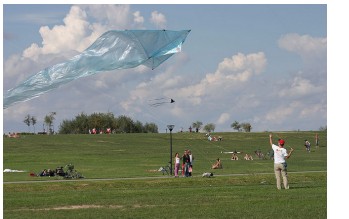

(3)

**Baseline:** a man is on the land with a big kite.

**CAAG:** a man is looking a kite in the sky.

**Ours:** a man is flying a kite on the grass land.

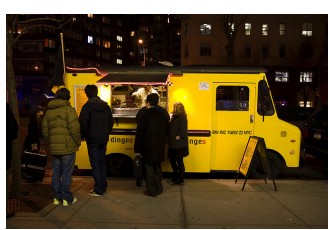

(4)

**Baseline:** a group of people are standing before a wagon .

**CAAG:** a group of people are standing before a truck for food.

**Ours:** a group of people ard standing before a yellow pie wagon for food.

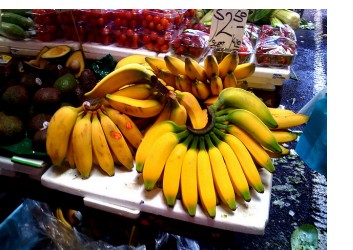

(5)

**Baseline:** some bananas are on the table with with.

**CAAG:** there are some bananas for sale on the gound with with .

**Ours:** there are some bananas for sale at the fruit stall.

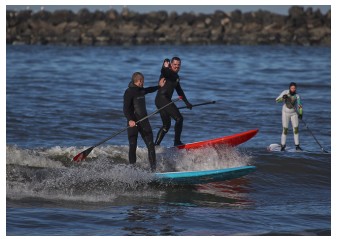

(6)

**Baseline:** there are several people standing on the water holding a rod.

CAAG: several people are standing in the lake holding oars.

**Ours:** several people are rowing in the lake with oars in their hands .

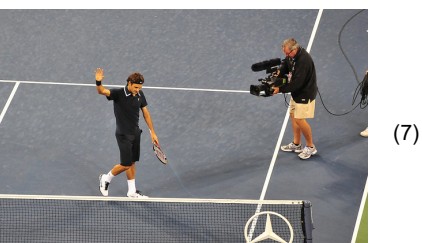

(7)

**Baseline:** a man is standing in the land and a man holds a camera.

**CAAG:** a tennis player is filmed by a man holding a camera.

**Ours:** a tennis player holding a racquet is filmed by a man holding a camera..

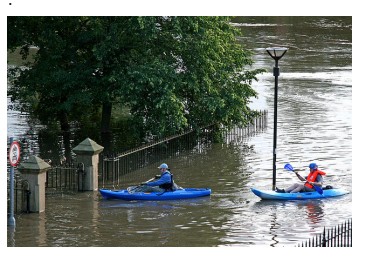

(8)

**Baseline:** several person and boats in the water.

**CAAG:** several people are sitting in the small boats.

**Ours:** a few people are sitting on small boat in the flooded streed next to a road lamp.

**Figure 5.** Visualization example of CSAN.

In the first image, the baseline method generated a correct caption sentence by listing the contents of the image. However, it failed to capture the relationship between the "person" and the "horse", specifically the action of "riding". In contrast, both CCAG and CSAN, which leverage context semantic information, accurately generated the term "riding". This clearly demonstrates the effectiveness of context semantic information in generating accurate and contextually relevant caption sentences. By considering the

contextual relationships and semantic information, CCAG and CSAN are able to produce captions that capture not just the objects in the image but also their relationships and actions, resulting in more comprehensive and accurate descriptions.

From the second to the sixth images, it can be seen that the image caption sentences generated by CSAN are more accurate and diverse than the two comparison methods. For example, in the second image, both the baseline method and CCAG generated grammatically correct caption sentences, but failed to accurately describe the image content. CSAA filtered out the erroneous information "standing", and generated a more accurate and diverse image caption based on context semantic information and visual information. In the third image, CCAG described the relationship "looking" between "man" and "kite" that the baseline method failed to give, but the relationship information was not accurate enough. CSAA filtered out inaccurate information and generated a more precise relationship "flying". In the fourth image, the baseline model could not use context semantic to obtain the information for "food", CAAG generated "truck" erroneously, and CSAN accurately solved the problems of both methods. In the fifth and sixth images, CSAN alleviated the adverse effects of erroneous information in context semantics in image captions, thus generating higher quality image description sentences.

In the seventh and eighth images, the caption sentences generated by CSAN accurately described the contents of the images, outperforming the two comparison methods. Specifically, neither the baseline method nor CAAG could predict the presence of the "racquet" in the seventh image and the "lamp" in the eighth image. However, CSAN successfully captured and accurately described these objects. This analysis highlights the superior performance of our proposed CSAN, which benefits from the comprehensive utilization of context semantic information, enabling more accurate and contextually relevant caption generation.

## 5. Conclusions

In this paper, we proposed the Attention-Aware (AA) module, extending the conventional attention mechanism, to filter out incorrect or irrelevant information. Furthermore, we proposed the Context Semantic Auxiliary Network (CSAN) for image captioning by applying AA to the decoder. Extensive comparative experiments and ablation analysis on the popular dataset MS COCO were conducted. The results demonstrated the superiority and general applicability of our proposed AA mechanism and CSAN.

**Author Contributions:** Conceptualization, J.L. and X.S.; methodology, J.L. and X.S.; software, J.L. and X.S.; validation, J.L.; formal analysis, X.S.; investigation, X.S.; resources, X.S.; data curation, J.L.; writing—original draft preparation, J.L.; writing—review and editing, X.S.; visualization, J.L.; supervision, X.S.; project administration, X.S.; funding acquisition, J.L. All authors have read and agreed to the published version of the manuscript.

**Funding:** This research was funded by the National Natural Science Foundation of China (Grant no. 62273142); the Research Foundation of the Education Bureau of Hunan Province, China (Grant no. 22A0490); the Hunan Enterprise Science and Technology Commissioner program (Grant no. 2021GK5074); and the science and technology innovation program of Hunan Province (Grant no. 2021GK2010).

**Data Availability Statement:** The dataset MS-COCO [40] and Flikr30K http://shannon.cs.illinois.edu/DenotationGraph/ (accessed on 12 June 2023) used in this study for experiments are openly available.

**Conflicts of Interest:** The authors declare no conflict of interest.

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
