# Peer review of "A Context Semantic Auxiliary Network for Image Captioning"

_information, doi:10.3390/info14070419_

Round 1
Reviewer 1 Report
The article presents a new mechanism to naturally generate image captioning. The authors used previous results from an LSTM language model and improved it with a proposal called CSAN, which is defined by two mechanisms, CSAA and CVAA. The authors performed different comparisons, indicating an improvement in the final performance.
The following sentence "However, these captioning methods predict the current word depending on the previously generated words, which resulting in a lack of context semantic information in the model." has not been substantiated. It is suggested that the authors could accompany their argumentation with empirical evidence.
In related work, the authors argue in relation to their work "AA mechanism to filter out the semantic information which is not needed for the generation of the image caption.". However, this is counterproductive, as this section presents other papers and related work, not a comparison between these. In summary, this section is very weak, and a very limited amount of related work has not been presented (without information).
In relation to the results, it is not clear how adequate or correct results are achieved because the article does not specify how they are evaluated. Despite this, the article is relevant, as only a major revision of the exposed sections is required to make it suitable for publication.
Others:
1) This study does not present any references. Thus, it is difficult to understand certain definitions and arguments.
2) Page 1. Line 27. Some terms are defined later in this paper (CAAG and RD).
3) Page 2. The terms RD and GTU have not yet been defined.
4) Page 2. Line 47. Fiure-->Figure
Reviewer 2 Report
This paper proposed Context Semantic Auxiliary Network (CSAN) to capture the context semantic information from the complete descriptive sentence. CSAN can improve the performance of the BUTD, RDN and AoA-Net on given metrics.
Weakness:
1. Computation efficiency matters. The network contains a Base Network and a CSAN in the generation process, meaning that it has two time-consuming steps. Number of parameters and inference time (e.g., FPS) should be further discussed.
2. The improvements over AoA-Net in Table 1 seems marginal. Deeper analysis should be made.
3. The literature review is far from complete. Some recent efforts in visual language reasoning that explore context information should be mentioned, such as Visual Abductive Reasoning, CVPR; Local-Global Context Aware Transformer for Language-Guided Video Segmentation, PAMI.
4. I suggest the authors report the re-implemented results of baselines (AoA-Net, BUTD, RDN, Adaptive) in table 1 and table 2 to make fair comparison.
5. “Our proposed method outperforms many compared methods on” cannot be viewed as a contribution. Conducting experiments is a way to verify the efficacy of your technique innovation; it is a means not the end.
The typos and errors are always here and there, such as "the visual features lacks" There are many typos and minor problems in this manuscript. For example, in line 27, what is CAAG? In line 29, the comma is misplaced. I suggest the authors go over the article carefully.
Reviewer 3 Report
This paper proposes a context-semantic auxiliary network model for image captioning. The contribution of the proposed approach is not well described. For example,
In Figure 1, it is not clear what the red- and green-highlighted words mean or what the difference is between the traditional method and the proposed method.
The proposed attention-aware AA module is missing in Figure 2, which presents the overview of the proposed approach.
- Figure 4 seems to be incomplete, please double check.
- What does the superscript index (e.g., h_t^1) mean in (2) and the following formulas? Please add brackets to avoid any confusion.
All reference indices display as "?" in the document. Please double check this issue.
There are a few typos in the document, for example.
- Line 55: "we propose" -> "We propose"
NA
Round 2
Reviewer 1 Report
The authors have resolved the points that had been raised. I have no further comments.
Author Response
Thank you very much for your suggestion, we refined the method description and the results presentation. The revisions to the manuscript are highlighted.
Reviewer 3 Report
The revision is fine, there are no further comments.
NA
Author Response
Thank you very much for the suggestion, we refined the manuscript, and the revisions to the manuscript are highlighted.